# Uptake of Ozenoxacin and Other Quinolones in Gram-Positive Bacteria

**DOI:** 10.3390/ijms222413363

**Published:** 2021-12-12

**Authors:** Yuly López, Laura Muñoz, Domingo Gargallo-Viola, Rafael Cantón, Jordi Vila, Ilonka Zsolt

**Affiliations:** 1Institute of Global Health of Barcelona, 08036 Barcelona, Spain; laura.munoz@isglobal.org; 2ABAC Therapeutics, 08950 Barcelona, Spain; dgargallo@abactherapeutics.com; 3Department of Clinical Microbiology, Hospital Universitario Ramón y Cajal & Instituto Ramón y Cajal de Investigación Sanitaria (IRYCIS), 28034 Madrid, Spain; rafael.canton@salud.madrid.org; 4CIBER Enfermedades Infecciosas, ISCIII, 28029 Madrid, Spain; 5Department of Clinical Microbiology, Hospital Clinic, School of Medicine, University of Barcelona, 08007 Barcelona, Spain; 6Ferrer International, 08029 Barcelona, Spain; izsolt@ferrer.com

**Keywords:** quinolone, uptake, ozenoxacin

## Abstract

The big problem of antimicrobial resistance is that it requires great efforts in the design of improved drugs which can quickly reach their target of action. Studies of antibiotic uptake and interaction with their target it is a key factor in this important challenge. We investigated the accumulation of ozenoxacin (OZN), moxifloxacin (MOX), levofloxacin (LVX), and ciprofloxacin (CIP) into the bacterial cells of 5 species, including *Staphylococcus aureus* (SA4-149), *Staphylococcus epidermidis* (SEP7602), *Streptococcus pyogenes* (SPY165), *Streptococcus agalactiae* (SAG146), and *Enterococcus faecium* (EF897) previously characterized.The concentration of quinolone uptake was estimated by agar disc-diffusion bioassay. Furthermore, we determined the inhibitory concentrations 50 (IC_50_) of OZN, MOX, LVX, and CIP against type II topoisomerases from *S. aureus*.The accumulation of OZN inside the bacterial cell was superior in comparison to MOX, LVX, and CIP in all tested species. The accumulation of OZN inside the bacterial cell was superior in comparison to MOX, LVX, and CIP in all tested species. The rapid penetration of OZN into the cell was reflected during the first minute of exposure with antibiotic values between 190 and 447 ng/mg (dry weight) of bacteria in all strains. Moreover, OZN showed the greatest inhibitory activity among the quinolones tested for both DNA gyrase and topoisomerase IV isolated from *S. aureus* with IC50 values of 10 and 0.5 mg/L, respectively. OZN intracellular concentration was significantly higher than that of MOX, LVX and CIP. All of these features may explain the higher in vitro activity of OZN compared to the other tested quinolones.

## 1. Introduction

A long list of compounds with antagonistic activity against bacteria have been described by various researchers worldwide [1]. Many of them are highly potent compounds isolated from natural sources [2]. However, most of them do not manage to advance in the phases of development of new drugs, being scarce new antibiotics against highly resistant bacteria. In general terms, there are numerous problems that an antibiotic must go through until it reaches its target of action at the intracellular level, including instability and binding to proteins in serum of the antibiotic reaching very low therapeutic concentrations at the site of infection until the low solubility and permeability of the drug not being able to penetrate the cell envelope of the bacteria, among others [3,4].

In this sense, most of the existing techniques for determining the cellular accumulation of compounds are based on the detection of fluorescence [5,6]. Specifically, studies based on the labelling of antibiotics with fluorescent probes to visualize the level of interaction with their target and intracellular accumulation, through high-resolution microscopy tools and unicellular microfluidics, have proven to be a simple, accessible, and with a great resolving power, being able to visualize the heterogeneity of a bacterial population against the action of an antibiotic [7,8,9].

Moreover, techniques based on LC-MS described by Zhou et al., 2015 [10] could be of great help in antibiotic accumulation studies due to the simplicity of the technique compared to other published methods and also the excellent detection power of antibiotics inside the bacterial cell. Finally, techniques such as bioassays based on disk diffusion methods of antibiotics in agar plates is a simple method of measuring the accumulation of the antibiotic inside the bacterial cell [11].

Quinolones are one of the largest classes of synthetic antibacterial agents with major clinical relevance, being one of the most frequently prescribed antibacterial agents in the world [12,13]. Initially, quinolones were mostly used in the treatment of Gram-negative infections. However, several modifications have been made to their basic structure to improve their pharmacokinetic properties and extend their antibacterial spectrum. In this way, quinolones have become effective against a wide variety of Gram-positive bacteria, including methicillin-susceptible *Staphylococcus aureus* (MSSA) and methicillin-resistant *S. aureus* (MRSA) [14,15]. However, over the past 50 years, the sporadic emergence of resistant strains has compromised the clinical usefulness of the quinolones currently available for the treatment of staphylococcal infections [16,17]. This upsurge in resistance has made it necessary to design new drugs for the treatment of infections that are caused by these resistant strains. In addition, serious side effects has been shown in several quinolones, which is why their prescription must take into account the benefit–risk ratio [18]. However, since ozenoxacin is a topical drug, it makes no systemic complications because its absorption is virtually null.

Presently, ozenoxacin (OZN) is the most recently-developed topical option for the treatment of skin infections. It belongs to a new generation of non-fluorinated quinolones and has shown great clinical benefit in two recent Phase III trials [19]. OZN has demonstrated excellent antibacterial activity against Gram-positive bacteria including strains which are resistant to other quinolones, low capacity to select resistant mutant strains and additionally, it has been shown that OZN activity is barely affected in strains with active efflux systems [20,21,22,23].

Quinolones can enter cells easily through porins in order to exert their antibacterial action via the inhibition of complex DNA-DNA gyrase and DNA-topoisomerase IV, enzymes which are both involved in bacterial DNA replication, transcription and repair [24]. However, the bactericidal effect of these antibiotics is related, at least in part, to the accumulation of reactive oxygen species (ROS) and the oxidative damage of several macromolecules [25,26]. The main mechanisms of resistance to quinolones in Gram-positive bacteria may be associated with the following: (i) chromosomal mutations in a specific region of the *gyrA* gene (encoding the DNA gyrase subunit A) and *grlA* gene (encoding the topoisomerase IV subunit A) called the Quinolone Resistance-Determining Regions (QRDR) [27,28]; (ii) chromosomal mutations leading to reduced drug accumulation by both decreased uptake associated with increased efflux; and (iii) Qnr-like determinants [29,30,31]. The expression of each of these mechanisms does not generally provide a high level of clinically significant resistance; however, they can accumulate and create highly quinolone-resistant strains that are difficult to treat in the clinical setting [25].

The ability of quinolones to enter bacterial cells contributes to their antibacterial potency. This potency is determined by their activity in stabilizing DNA complexes with DNA gyrase and topoisomerase IV and by their ability to permeate cell membranes and avoid efflux in order to reach these targets [26,32]. The latest generation of quinolones, including OZN, are compounds with more closely balanced activities against type II topoisomerases and with substantially lower frequencies of resistance in vitro [33]. In this sense, OZN exhibits strong inhibitory activity at low concentrations which might be due to its dual target of action [33,34]. Nevertheless, the concentration of OZN accumulated by Gram-positive cocci and its relationship with activity has not been clearly examined.

In this study, the investigation focused on the accumulation of OZN in Gram-positive cocci compared with other quinolones and the inhibitory capacity of OZN and other quinolones to type II topoisomerases of *S. aureus*.

## 2. Results and Discussion

### 2.1. Quinolone Uptake

The results regarding the accumulation of OZN, MOX, LVX, and CIP inside bacterial cells of different species of Gram-positive bacteria are shown in Figure 1. We predominantly focused on the main pathogens associated with skin infections, such as *S. aureus*, *S. epidermidis,* and *S pyogenes*, but strains of *S. agalactiae* and *E. faecium* were also included.

It is important to mention that there is a significant number of publications which describe different methods for measuring the accumulation of quinolones in Gram-positive bacteria [5,11]. These methods are mainly based on fluorescence techniques measuring the appropriate excitation and emission wavelength for each antibiotic. However, in our study, OZN showed poor fluorescence emission according to data obtained by compared with other tested quinolones. Therefore, in our study the methodology used to measure quinolone accumulation was the bioassay technique according to published by Cazedey and Salgado, 2011 with some modifications [35].

According to our results, there was a marked difference between the cumulative quinolone concentration values. Higher levels of OZN accumulation were observed in comparison with that of MOX, LVX, and CIP in all tested species. The rapid penetration of OZN into the cell was reflected during the first minute of exposure to the antibiotic, observing accumulation values of between 190 and 447 ng/mg (dry weight) of bacteria in all strains. MOX was accumulated in high concentrations in the first minute of exposure inside the bacterial cell of *E. faecium* and *S. epidermidis* in comparison with LEV and CIP (values between 168 and 312 ng/mg (dry weight) of bacteria, respectively). However, the accumulation values of LEV and CIP were lower in the first minute of exposure, being less than 50 ng/mg (dry weight) of bacteria in all tested species (except in *S. pyogenes*).

Ozenoxacin’s physical-chemical characteristics prompt a rapid penetration into the bacterial cells, achieving a quick interaction with their targets of action which translates into the strong activity shown by OZN against Gram-positive bacteria.

Previous reports have shown that the hydrophobicity of quinolones influences the final concentrations of antibiotic accumulated by bacteria [11,36]. For Gram-negative bacteria, it has been reported that the more hydrophilic the molecule, the higher the concentration accumulated within the bacterium. However, generally speaking, for Gram-positive bacteria the opposite is true, i.e., the higher the hydrophobicity, the greater the concentration accumulated [36]. In this sense, OZN and MOX are the most strongly hydrophobic compounds (logP value of 2.76 and 0.01, respectively), followed by LVX and CIP (logP values of −0.01 and −0.57, respectively), the latter being considered more hydrophilic [23,37]. Thus, the influence of the higher hydrophobicity of OZN (according to data logP) could favor to the higher accumulation within the bacterial strains throughout the experiment, followed by MOX, LVX, and CIP.

On the other hand, the molecular weight of an antibiotic has a significant relationship with the accumulation of these compounds. In prior studies it has been described that fluoroquinolones with the lowest molecular weight accumulate to the highest concentrations while fluoroquinolones with high molecular weights accumulate to the lowest concentrations [38]. In our study, we did not encounter this relationship, as ciprofloxacin which comparatively has the lowest molecular weight (331.347 g/mol) did not achieve higher accumulated concentrations. Furthermore, MOX, which has the highest molecular weight (401.438 g/mol) showed a lower accumulated concentration in comparison with OZN, but higher with respect to LEV and CIP. The molecular weight of OZN is 363.41 g/mol, which is similar to that of LVX (361 g/mol) and CIP (331 g/mol). However, the accumulated concentration of OZN was higher in all bacterial strains of the study, followed by MOX which has higher molecular weight. In the study of Piddock et al. [38], there were also exceptions to this principle related to molecular weight which leads to the idea that other factors could influence the accumulation rate. In this sense, the study of Cama et al., 2016 [39], comment that the presence of fluorine atom on the aromatic ring of some quinolones could be increase the lipophilicity and the permeability of antibiotic generating a greater accumulation inside the bacterial cell. However, tin their research showed that fluorine atom produces the opposite effect, reducing the accumulation of antibiotics under the presence of chemical substituents of fluorine.

Seeking to explain the rapid accumulation of OZN, we looked into in previous work from our laboratory and observed that classical transporters present in Gram-positive bacteria, such as NorA and MepA, do not affect OZN activity. Therefore, despite affecting other quinolones, the constitutive expression of these efflux pumps could not transport OZN outside the cell, probably due to the hydrophobicity of C-8 and the bulkiness of C-7 [23] (Figure 2), allowing for high accumulation values of OZN.

### 2.2. Inhibition of DNA Gyrase and Topoisomerase IV Assays

One of the mechanisms responsible for resistance to quinolones are point mutations in the genes encoding the target enzymes. On the whole, the mutations occur in the QRDR regions of DNA gyrase and topoisomerase IV. In the case of *gyrA* and *parC* genes (*grlA* in *S. aureus*), the mutations are usually located at the 5′-end and in the midregion for *gyrB* and *parE* genes (*grlB* in *S. aureus*) [40]. Several researchers describe that a mutation in the more sensitive enzyme (primary target) results in an increase in the MIC of a quinolone, whereas a mutation in the less sensitive enzyme (secondary target) generally causes resistance only in the presence of resistance mutations in the primary target [41]. However, a quinolone with similar affinities for both targets is largely unaffected by a mutation in one of the enzymes, with several mutations in both enzymes being necessary for resistance to develop [28,42].

According to the results obtained in this study, of all the tested quinolones, OZN showed the greatest inhibitory activity for both DNA gyrase and topoisomerase IV isolated from *S. aureus* among the tested quinolones (Figure 3 and Table 1).

The inhibitory concentration 50 (IC_50_) values of OZN against DNA gyrase and topoisomerase IV of *S. aureus* were 10 and 0.5 mg/L, respectively, whereas the inhibitory activity against DNA gyrase of MOX was 5 times higher than OZN, with and IC_50_ value of 56 mg/L and slightly higher against topoisomerase IV, with an IC_50_ of 0.95 mg/L. In the case of LVX and CIP, higher IC_50_ values against DNA gyrase were observed with values of >100 and 98 mg/L, respectively. Nevertheless, LVX showed good activity against topoisomerase IV, with IC_50_ value of 0.45 mg/L, when compared with CIP (1.3 mg/L).

This substantial inhibitory activity for both OZN enzymes was described by Yamakawa et al. [33]. Previous studies with T-3912 (now OZN) have shown IC_50_ values for both, topoisomerase IV and DNA gyrase, to be the lowest among the tested quinolones (0.617 and 4.5 mg/L, respectively). The difference in respect to the IC_50_ value of DNA gyrase observed in our study is probably due to differences in methodologies. In our study, we used topoisomerase available commercially. However, in the study of Yamakawa, the DNA topoisomerase IV (grlA and grlB) and DNA gyrase (gyrA and gyrB) were isolated from *S. aureus* SA113. Both enzymes were cloned and expressed and in *E. coli* DH5α.

Furthermore, Yamakawa supports the dual targeting of T-3912 (OZN) because the activity of T-3912 (OZN) was not influenced by *grlA* mutations in *S. aureus*, whereas it increased two-fold for a *gyrA* mutation, suggesting that it, too, has gyrase as its primary target in *S. aureus*. From our group, similar results were published, demonstrating that the MIC of OZN was considerably lower in strains of *S. aureus* regardless of the number of mutations in the *gyrA* and/or *parC* genes in comparison with other tested quinolones [43].

In general terms and owing to extensive research efforts, the quinolones of today have better activity with better clinical efficacy, reduced resistance selection, and safety. In this sense, the des-fluoro (6) quinolone group, that lacks the classical C-6 fluorine characteristic of the previous generation of fluoroquinolones, has significant differences in its basic structure [44].

Garenoxacin together with OZN, nemonoxacin, WCK-1734, PGE 9262932 and PGE 9509924 are the main representatives of this group and have substantial differences in the C-7 and C-8 substituents [45,46,47,48]. Position 7 includes very bulky elements, such as pyrrolidine or piperazine, both considered to directly interact with DNA gyrase, or topoisomerase IV, greatly influencing the potency, spectrum, and pharmacokinetics. Moreover, it appears to confer protection from the efflux exporter proteins of bacteria [49].

On the other hand, the elements in position C-8 such as methyl (in the case of OZN and WCK-1734) or methoxy and/or difluoromethoxy (in the case of garenoxacin), have been shown to improve bacteriostatic and bactericidal activity and decrease the selection of resistant mutants. Additionally, the hydrophobicity of this substituent could favor a greater accumulation of the antibiotic inside the bacterial cell [50].

## 3. Materials and Methods

### 3.1. Quinolone Uptake

#### 3.1.1. Bacterial Strains

Five strains with genetic profiles in the QRDR region previously well-characterized by PCR and sequencing were analyzed in this study (Table 2). The strains were selected from a previous study and obtained from the Clinical Microbiology Laboratory at the Hospital Clinic in Barcelona, Spain [43].

#### 3.1.2. Antibacterial Agents

The quinolones used in this study were: ozenoxacin (OZN, Ferrer Internacional S.A. Barcelona, Spain), moxifloxacin (MOX), levofloxacin (LVX) and ciprofloxacin (CIP) obtained from Sigma-Aldrich (St. Louis, MO, USA).

#### 3.1.3. Accumulation Assay

Quinolone uptake was assayed by the method of Giraud et al., 2000 [51], with some modification. The bacteria were grown at 37 °C in Mueller-Hinton (MH) broth (Condalab, Madrid, Spain) with shaking set at 180 rpm until an optical density [OD] of 0.7 at 600 nm was reached. These cultures were centrifuged and resuspended in 1× phosphate-buffered saline (PBS) (pH 7.2). The resuspended cells were equilibrated for 10 min at 37 °C in sterile falcon tubes (Deltalab, Barcelona, Spain). The quinolone to be evaluated was added to a final concentration of 20 mg/L. After the addition of quinolone, aliquots of 0.5 mL (in duplicate, one set of tubes was used to calculate the dry weight of bacteria) were removed at different time intervals until 5 min was reached, including an aliquot without exposure to the antibiotic to be used as a control, considering it as time 0. Each aliquot was immediately diluted in 1 mL of ice-cold 1xPBS buffer and was centrifuged at 13,500 rpm for 8 min at 4 °C. The pellet obtained was washed three times in 1 mL of ice-cold 1xPBS buffer to eliminate any traces of the extracellular antibiotic. Finally, the pellet was resuspended in 1 mL of glycine hydrochloride 0.1 M (pH 3.0) for at least 15 h in the dark at room temperature. Once the had time elapsed, the samples were centrifuged at 13,500 rpm for 10 min at 4 °C. The quinolone concentration in the supernatant was estimated by bioassay according to published by Cazedey and Salgado, 2011 with some modifications [35].

#### 3.1.4. Preparation of Bioassay Plates

Two sets of plates were prepared for each analysis, depending on the quinolone concentration to be quantified. Each plate consists of two layers of MH agar (Condalab, Madrid, Spain). The first layer consists of 17.5 mL of MH agar, and this volume was deposited inside the sterile petri and solidified at room temperature for 10 min. The second layer contains 12.5 mL of MH agar with bacteria. This layer was prepared by adding to 100 mL of MH agar at a temperature of 45 °C, 1 mL of bacterial culture of the reference strain at a concentration of ~10^8^ CFU/mL (McFarland 0.5). After the agar hardened, the plates were stored at 4 °C and were used on the day they were prepared or on the following day. The reference strains used were *E. coli* ATCC 25922 and *S. aureus* ATCC 29213.

#### 3.1.5. Bioassay Procedure

For each accumulation assay, six Eppendorf tubes (from t0 to t5) were obtained containing the supernatant with the quinolone to be quantified. From each tube, 25 µL of the supernatant was used to impregnate sterile paper discs (6 mm diameter Whatman) and were dried to room temperature under sterile conditions. Each disc was placed in the respective set of bioassay plates (6 discs per plate). The plates were incubated for 24 h at 37 °C. The inhibition zone around the disk was measured. After standardization of experiment. Each bioassay was realized one-time.

#### 3.1.6. Standard Curve

To determine the linearity of response to quinolone, a standard curve was performed. For this, serial dilutions of each antimicrobial to be evaluated were prepared in concentrations of quinolone from 0.0156 to 1024 mg/L diluted in sterile distilled water (except OZN, which was diluted in 1 N NaOH). Then, employing the procedure previously explained, the sterile discs were impregnated with 20 µL of each concentration of antibiotics and were deposited on the bioassay plates.

To calculate the cytoplasmic concentration of quinolone accumulated in any of the strains analyzed in the study, the values of the halos generated by the samples were extrapolated on the standard curve. The tests were carried out in duplicate and the results obtained were expressed in nanograms (ng) of quinolone per milligram (mg) of dry weight of the bacteria.

### 3.2. Inhibition of DNA Gyrase and Topoisomerase IV Assays

#### 3.2.1. Enzymatic Assay

The enzymatic assay was carried using commercial topoisomerase type II as previously described by Alt et al. [52] with some modifications. Briefly, three units of the enzyme (gyrase or topo IV, Inspiralis. Norwich, UK) isolated from *S. aureus* converts 0.5 mg of relaxed pBR322 DNA to the supercoiled form (gyrase) or decatenates 200 ng of kinetoplast DNA (topo IV). The enzymatic assay was performed by incubation for 30 min at 37 °C in a total reaction volume of 30 µL. Standard reaction mixtures for the gyrase supercoiling assays contained 35 mM Tris-HCl (pH 7.5), 24 mM KCl, 700 mM K-Glu, 4 mM MgCl2, 2 mM DTT, 1.8 mM spermidine, 1 mM ATP, 6.5% (*w*/*v*) glycerol, and 0.1 mg/mL albumin. Topo IV activity was measured by using a decatenation assay that monitored the ATP-dependent unlinking of DNA minicircles from kDNA containing 40 mM HEPES-KOH (pH 7.5), 100 mM K-Glu, 10 mM magnesium acetate, 10 mM DTT, 1 mM ATP, and 50 mg/mL albumin. The reactions were stopped by adding 30 µL of stop buffer [Chloroform/Isoamyl alcohol (24:1) y 8 µL of Bromophenol Blue]. Then, 25 µL of the aqueous phase of each sample was analyzed on 1% agarose gels for 4 h at 80 V in 1XTAE (40 mM Tris, 20 mM acetic acid, and 1mM EDTA at pH 8.0) and visualized after staining with SYBR^®^ Gold Nucleic Acid stain (Molecular Probes Inc., PoortGebouw, The Netherlands) under UV light. For each analysis, a negative control that contained the buffer tampon and plasmid pBR322/kinetoplast, and a positive control that contained 1 U of enzyme (gyrase or topo IV) were used, and finally, four reactions with different concentrations of quinolones were analyzed.

#### 3.2.2. Gel Analysis and IC50 Values

The analysis of gel electrophoresis bands was performed using G:BOX software (Chemi XT4, Syngene, GeneTools 4.3.14) to determine the inhibitory concentration 50 (IC_50_) for each antibiotic. The IC_50_ was defined as the concentration causing 50% inhibition of the supercoiling or the decatenation reaction, as seen with the drug-free controls.

## 4. Conclusions

The strong inhibitory activity of ozenoxacin at low concentrations might be explained by its rapid penetration of the bacterial cell in the first minute after exposure, reaching high intrabacterial concentrations compared with other quinolones in all studied microorganisms, among them *S. aureus* and *S. pyogenes*, which are causal agents of impetigo. Furthermore, this fast penetration of OZN allows for a quick interaction with the targets of action, inhibiting DNA gyrase supercoiling activity and topoisomerase IV decatenation simultaneously. All of these features may explain the higher in vitro activity of OZN compared to the other tested quinolones.

## Figures and Tables

**Figure 1 ijms-22-13363-f001:**
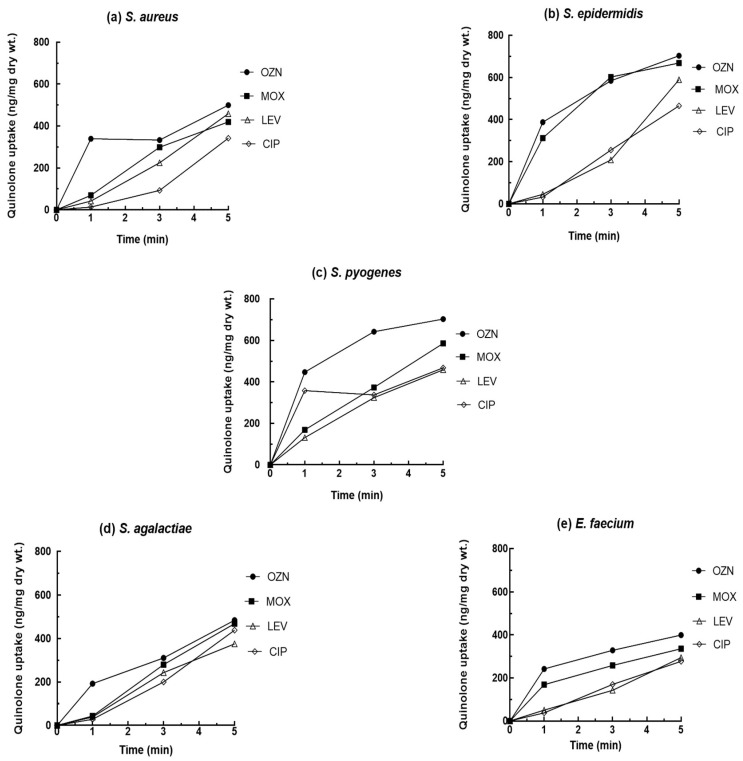
Accumulation of 20 mg/L of quinolone for *S. aureus* (**a**), *S. epidermidis* (**b**), *S. pyogenes* (**c**), *S. agalactiae* (**d**), and *E. faecium* (**e**).

**Figure 2 ijms-22-13363-f002:**
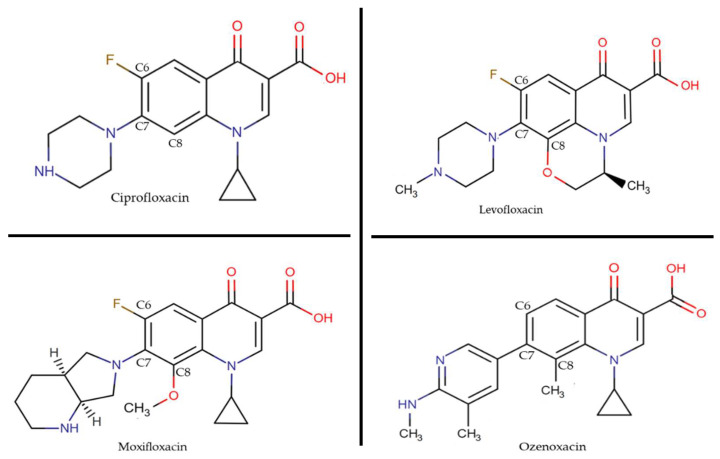
Chemical structures of quinolones used in this study.

**Figure 3 ijms-22-13363-f003:**
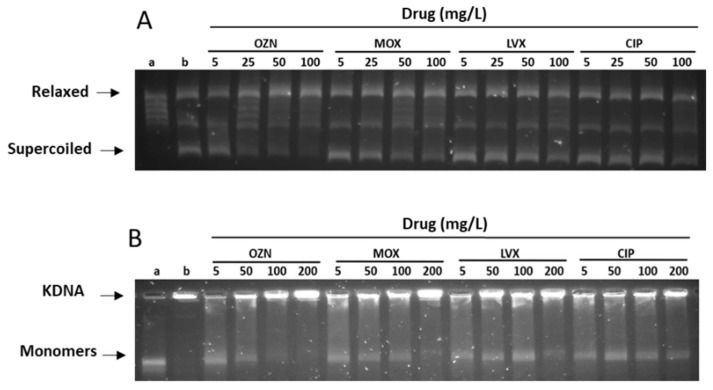
Quinolone inhibition of DNA supercoiling and DNA decatenation in the presence of different concentrations of quinolones (ciprofloxacin (CIP), moxifloxacin (MOX), levofloxacin (LVX), and ciprofloxacin (CIP). (**A**), Inhibition of DNA supercoiling by *S. aureus* gyrase. (**B**), Inhibition of DNA decatenation by *S. aureus* topoisomerase IV. Lanes a and b showed relaxed and supercoiled pBR322 DNA, respectively in the assay DNA supercoiling or KDNA and monomers in the assay of DNA decatenation.

**Table 1 ijms-22-13363-t001:** Inhibitory activity of ozenoxacin, moxifloxacin, levofloxacin, and ciprofloxacin against DNA gyrase and topoisomerase IV obtained from *S. aureus*.

Antibiotic	*S. aureus* IC_50_ (mg/L)	
	DNA Gyrase	Topoisomerase IV
Ozenoxacin	10	0.5
Moxifloxacin	56	0.95
Levofloxacin	>100	0.45
Ciprofloxacin	98	1.3

**Table 2 ijms-22-13363-t002:** Characteristics of bacterial strains used in this study.

Strains	Minimum Inhibitory Concentration (MIC) (mg/L)				Mutation in QRDR
	OZN −R/+R ^1^	MOX −R/+R	LVX −R/+R	CIP −R/+R	
*S. aureus* (4-149)	0.0039/0.0039	0.25/0.25	0.25/0.25	0.38/0.38	WM ^2^
*S. epidermidis* (7602)	0.03/0.008	0.06/0.06	0.5/0.25	1/0.5	WM
*S. pyogenes* (165)	0.12/0.12	1/0.06	16/0.06	4/2	WM
*S. agalactiae* (146)	0.06/0.03	0.25/0.25	2/1	0.25/0.25	WM
*E. faecium* (897)	0.25/0.25	0.5/0.5	2/2	1/1	Ser80Ile (parC)

^1^ Reserpine (20 mg/L), ^2^ Without mutation in QRDR.

## Data Availability

Not applicable.

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
