# Peer review of "Uptake of Ozenoxacin and Other Quinolones in Gram-Positive Bacteria"

_ijms, 2021, doi:10.3390/ijms222413363_

Round 1
Reviewer 1 Report
This paper reports on the intracellular accumulation of quinolones in gram-positive bacteria. The paper could be worth publication in IJMS after the following issues have been addressed.
Abstract:
The abstract would benefit from an introductory statement on the field of research under investigation, i.e. antibiotic therapy
please define OZN.
What do you mean with 447 ng/mg? Please report concentration in molarity or g/L or specify otherwise
What do you mean with “OZN showed 23 the greatest inhibitory activity among the quinolones tested for both DNA gyrase and topoisomerase” are these in vitro measurements against purified gyros/topoisomerase?
Introduction: very well written and organised, however, it needs two major improvements:
1) it would greatly benefit from an introductory statement on the field of research under investigation before going straight into quinolones
2) it needs explaining what is the state-of-the-art of the investigation of intracellular antibiotic and, more generally, molecular accumulation in bacteria and citing at least the following key papers (beyond refs 23-24 reported in the results section):
https://pubmed.ncbi.nlm.nih.gov/10381102/
https://pubmed.ncbi.nlm.nih.gov/33596653/
https://royalsocietypublishing.org/doi/full/10.1098/rstb.2018.0442
https://pubs.acs.org/doi/10.1021/ac504880r
https://pubs.rsc.org/en/content/articlelanding/2020/cb/d0cb00118j
https://doi.org/10.1016/j.cbpa.2018.05.005
https://www.biorxiv.org/content/10.1101/2021.10.18.464851v1
https://pubmed.ncbi.nlm.nih.gov/18997824/
http://dx.doi.org/10.1039/D0LC00242A
Results
“We predominantly focused on the main pathogens associated with skin infections, such as S. aureus,
S. epidermidis and S pyogenes, but strains of S. agalactiae and E. faecium were also included”
How closely related are these species?
“Therefore, in our study the methodology used to measure quinolone accumulation
was the bioassay technique.”
This needs either explaining or a reference
“there was an observably significant difference”
how was significance statistically tested and where are the results of these tests reported?
“MOX accumulated higher concentration than LVX and CIP”
This sentence does not make any sense please rephrase
Figure 1: please report error bars and number of biological and technical replicates
“However, generally speaking, for Gram-positive bacteria the opposite is true”
Please provide a reference and a mechanistic explanation for this
“In prior studies it has been described that
fluoroquinolones with the lowest molecular weight accumulate to the highest concentra-
tions while fluoroquinolones with high molecular weights accumulate to the lowest con-
centrations [27]”
Please compare your data to other previous in vitro findings, e.g. https://www.nature.com/articles/srep32824
Author Response
His paper reports on the intracellular accumulation of quinolones in gram-positive bacteria. The paper could be worth publication in IJMS after the following issues have been addressed.
Abstract:
- The abstract would benefit from an introductory statement on the field of research under investigation, i.e. antibiotic therapy. Please define OZN.
-We add that “The big problem of antimicrobial resistance requires great efforts in the design of improved drugs which can quickly reach their target of action. Studies of antibiotic uptake and interaction with their target it is a key factor in this important challenge. We investigated the accumulation of ozenoxacin (OZN)”
- What do you mean with 447 ng/mg? Please report concentration in molarity or g/L or specify otherwise.
- We used in our data expression (ng/mg) to obtain a better specificity of nanograms (ng) of quinolone considering the milligrams (mg) of dry weight of the bacteria. These are the units normally used for quinolone accumulation.
- What do you mean with “OZN showed the greatest inhibitory activity among the quinolones tested for both DNA gyrase and topoisomerase” are these in vitro measurements against purified gyros/topoisomerase?
- Yes, we worked with purified topoisomerase type II purchased from Inspiralis (Norwich, UK) and the activity of both enzymes was tested using specific assays as mentioned in the manuscript.
Introduction: very well written and organised, however, it needs two major improvements:
- it would greatly benefit from an introductory statement on the field of research under investigation before going straight into quinolones.
- it needs explaining what is the state-of-the-art of the investigation of intracellular antibiotics and, more generally, molecular accumulation in bacteria and citing at least the following key papers (beyond refs 23-24 reported in the results section):
Thanks for these comments, in our introduction, we wanted to explain the necessity of new antibiotics due to the big problem of resistance to quinolone and toxicity of them until a new drug called ozenoxacin that have a very good antibacterial activity due to rapid penetration of OZN into the cell and the strong inhibitory activity against both DNA gyrase and topoisomerase IV (results of this paper). However, and according to the comment in point 1 and 2 we add a new paragraph in the new version of paper.
“A long list of compounds with antagonistic activity against bacteria have been described by various researchers worldwide [1]. Many of them are highly potent compounds isolated from natural sources [2]. However, most of them do not manage to advance in the phases of development of new drugs, being scarce new antibiotics against highly resistant bacteria. In general terms, there are numerous problems that an antibiotic must go through until it reaches its target of action at the intracellular level, including instability and binding to proteins in serum of the antibiotic reaching very low therapeutic concentrations at the site of infection until the low solubility and permeability of the drug not being able to penetrate the cell envelope of the bacteria, among others [3,4].
On the other hand, most of the existing techniques for determining the cellular accumulation of compounds are based on the detection of fluorescence [5,6]. However, this method is not suitable for compounds that lack fluorescent remains and also have bulky substituents in their structure. In this sense, techniques such as bioassays based on the based-on disk diffusion methods of antibiotics in agar plates is a simple method of measuring the accumulation of the antibiotic inside the bacterial cell [7]. Moreover, techniques based on LC-MS described by Zhou et al, 2015 [8] could be of great help in antibiotic accumulation studies due to the simplicity of the technique compared to other published methods and also the excellent detection power of anti-biotics inside the bacterial cell.
- it needs explaining what is the state-of-the-art of the investigation of intracellular antibiotics and, more generally, molecular accumulation in bacteria and citing at least the following key papers (beyond refs 23-24 reported in the results section):
Results
- “We predominantly focused on the main pathogens associated with skin infections, such as S. aureus, S. epidermidis and S pyogenes, but strains of S. agalactiae and E. faecium were also included” How closely related are these species?
-We have included the S. agalactiae and E. faecium in this study, because both strains are related (a lesser degree) as etiological agents in skin infections, such as cellulitis, abscesses, foot infection and necrotizing fasciitis (10.1128 / microbiolspec.GPP3-0007-2018, 10.4103 / 0974-777X.145253). Moreover, we have wanted to include strains with higher MIC values to quinolone including OZN (in the case of E. faecium).
- “Therefore, in our study, the methodology used to measure quinolone accumulation was the bioassay technique.” This needs either explaining or a reference
-Ozenoxacin, apart from other quinolones, has low emission of fluorescence. Unfortunately, this information was not previously published. However, we were able to assess and quantify this low emission of fluorescence by OZN in the school of Pharmacy at the University of Barcelona. For this reason, we decided to carry out the bioassay technique, after standardization in the laboratory based on the protocol described by Cazedey 2011. This method depends on upon diffusion of the antibiotic solution through a solidified agar layer (or double-layer) in a Petri plate. Therefore, in our study, we have tested different volumes of agar to obtain inhibition halos with OZN against S. aureus present in the upper layer of agar. We add the new paragraph and reference in the new version of paper.” However, in our study, OZN showed poor fluorescence emission according to data obtained by compared with other tested quinolones (data not shown). Therefore, in our study, the methodology used to measure quinolone accumulation was the bioassay technique according to published by Cazedey and Salgado, 2011 with some modifications [26]”.
- “there was an observably significant difference” how was significance statistically tested and where are the results of these tests reported?
-We're sorry, but we have not applied statistical tests in our results. We add a new sentence to avoid confusion in the new version of paper "According to our results, there was a marked difference between the cumulative quinolone concentration values"
- “MOX accumulated higher concentration than LVX and CIP” This sentence does not make any sense please rephrase
- Thanks for this comment, we add a new paragraph in the new version of paper. "MOX was accumulated in high concentrations in the first minute of exposure inside the bacterial cell of E. faecium and S. epidermidis in comparison with LEV and CIP (values between 168 and 312 ng/mg (dry weight) of bacteria, respectively)"
- Figure 1: please report error bars and number of biological and technical replicates
- We're sorry, but it is impossible to address this information. The accumulation assays by bioassay needs extensive standardization. Unfortunately, all the tests have some methodological differences, for example, the volume of aliquot to impregnate the disc, the preparation of the plates for bioassay, among others. Unfortunately, the results are not comparable. However, if you prefer, we can add as supplementary material the standard curve with its respective graphs.
- “However, generally speaking, for Gram-positive bacteria the opposite is true”
Please provide a reference and a mechanistic explanation for this
- We add the reference in the new version of paper.
- “In prior studies it has been described that fluoroquinolones with the lowest molecular weight accumulate to the highest concentrations while fluoroquinolones with high molecular weights accumulate to the lowest concentrations [27]”
Please compare your data to other previous in vitro findings, e.g. https://www.nature.com/articles/srep32824
--The study of Cama et al, 2016 presents a methodology quite different from that of our study. In their work, they have investigated the differences in the permeability of four fluoroquinolones over a range of pH values, considering the increase of lipophilicity as indicator of permeability increase. However, they have also made a relation between the structure of chemical substituents of quinolones investigated and the permeability, observing how are these subtle differences in the structure of substituents could increase the range of permeability coefficients. In sense, OZN have similar substituent in C-7 to enrofloxacin, and this chemical substituent could be conferring a higher permeability value and with it a greater accumulation. On the other hand, the researchers comment that the presence of fluorine atom on the aromatic ring is expected to increase the lipophilicity and the permeability. However, in the study they have shown that fluorine atoms in the aliphatic produce the opposite effect. Concerning our study, we compared OZN with 3 fluorinated quinolones, that according to the results of Cama et al, this kind of antibiotics have a reduced lipophilic activity and lower permeability.
We add the paragraph “In this sense, the study of Cama et al, 2016 [30], comment that the presence of fluorine atom on the aromatic ring of some quinolones could be increase the lipophilicity and the permeability of antibiotic generating a greater accumulation inside the bacterial cell. However, in their research showed that fluorine atom produces the opposite effect, reducing the accumulation of antibiotics under the presence of chemical substituents of fluorine.)
Reviewer 2 Report
The article Uptake of ozenoxacin and other quinolones in Gram-positive bacteria, by López and collaborators, is part of a series of research designed to provide new therapeutic solutions that can be used to combat microbial resistance.
Quinolones are an important class of antimicrobial drugs, but lately they have attracted attention due to serious side effects, which is why their prescription must take into account the benefit-risk ratio. It would have been good if the authors also mentioned this aspect in the introduction.
Also, the study, although is well done, does not amount to the value it should have in order to be published in a prestigious journal such as Int. J. Mol. Sci . For example, molecular docking studies would have been necessary. Also, the conclusions are the ones expected, demonstrated in other articles published by López and his collaborators.
For the reasons stated, I do not agree with the publication of the article in its current form.
Author Response
The article Uptake of ozenoxacin and other quinolones in Gram-positive bacteria, by López and collaborators, is part of a series of research designed to provide new therapeutic solutions that can be used to combat microbial resistance.
Quinolones are an important class of antimicrobial drugs, but lately they have attracted attention due to serious side effects, which is why their prescription must take into account the benefit-risk ratio. It would have been good if the authors also mentioned this aspect in the introduction.
Thanks for this comment for which we fully agree. We have added this consideration into the introduction section. However, since ozenoxacin is a topical drug, it makes no sense to consider possible systemic complications because its absorption is virtually nil. Precisely this is an advantage over other quinolones.
Also, the study, although is well done, does not amount to the value it should have in order to be published in a prestigious journal such as Int. J. Mol. Sci . For example, molecular docking studies would have been necessary. Also, the conclusions are the ones expected, demonstrated in other articles published by López and his collaborators.
The reviewer mentioned that “the conclusions are the ones expected, demonstrated in other articles published by López and his collaborators”. We do not agree with this sentence since we think in the conclusions section it is stated the additional value of our results which helps to explain why ozenoxacin shows a very good activity against Gram positive bacteria. In fact, this great activity compared to other quinolones is explicated by two factors, a rapid accumulation of ozenoxacin plus a low inhibitory concentration (IC50) for the two protein targets DNA gyrase and topoisomerase IV, both factors reported in this manuscript.
Round 2
Reviewer 1 Report
The authors have addressed most of my comments. However, the introduction on antibiotic accumulation still needs expanding on existing literature, I have indicated a long list of examples in the previous report which I am reporting again here:
https://pubmed.ncbi.nlm.nih.gov/10381102/
https://pubmed.ncbi.nlm.nih.gov/33596653/
https://royalsocietypublishing.org/doi/full/10.1098/rstb.2018.0442
https://pubs.acs.org/doi/10.1021/ac504880r
https://pubs.rsc.org/en/content/articlelanding/2020/cb/d0cb00118j
https://doi.org/10.1016/j.cbpa.2018.05.005
https://www.biorxiv.org/content/10.1101/2021.10.18.464851v1
https://pubmed.ncbi.nlm.nih.gov/18997824/
http://dx.doi.org/10.1039/D0LC00242A
The statement "However, this method is not suitable for compounds that lack fluorescent " is incorrect, fluorescent antibiotics can be generated for most antibiotic classes, see above
Author Response
Thanks to you for all your suggestions on this paper.

Reviewer 2 Report
I believe that by bringing additions to the article, its scientific quality has increased and it can be published in its current form.
I thank the authors for the additions and comments made.
Author Response
1. The authors have addressed most of my comments. However, the introduction on antibiotic accumulation still needs expanding on existing literature, I have indicated a long list of examples in the previous report which I am reporting again here:
*Sorry, we added some of the suggested references, but we haven't detailed it in the answer. In this new version, we have include a little paragraph about fluorescence:
“Specifically, studies based on the labelling of antibiotics with fluorescent probes in order to visualize the level of interaction with their target and intracellular accumulation, through high-resolution microscopy tools and unicellular microfluidics, have proven to be a simple, accessible and with a great resolving power, being able to visualize the heterogeneity of a bacterial population against the action of an antibiotic. [7–9]”
We had included these papers:
1. https://pubmed.ncbi.nlm.nih.gov/10381102/
-We add this reference in the introduction (ref.5)
2. https://pubmed.ncbi.nlm.nih.gov/33596653/
-We add this reference in the introduction (ref.4)
3. https://pubs.acs.org/doi/10.1021/ac504880r
-We add this reference in the introduction (ref.10)
4. https://doi.org/10.1016/j.cbpa.2018.05.005
-We add this reference in the introduction (ref.6)
https://pubs.rsc.org/en/content/articlelanding/2020/cb/d0cb00118j
-We add this reference in the introduction (ref.7)
https://www.biorxiv.org/content/10.1101/2021.10.18.464851v1
-We add this reference in the introduction (ref.9)
http://dx.doi.org/10.1039/D0LC00242A
-We add this reference in the introduction (ref.8)
2. The statement "However, this method is not suitable for compounds that lack fluorescent" is incorrect, fluorescent antibiotics can be generated for most antibiotic classes, see above
-We removed this statement.

Round 3
Reviewer 1 Report
The authors have now addressed all the outstanding issues and the manuscript is suitable for publication